# Approximate Heavily-Constrained Learning with Lagrange Multiplier Models

**Harikrishna Narasimhan, Andrew Cotter, Yichen Zhou**
Google Research, USA
{hnarasimhan,acotter,yichenzhou}@google.com

**Serena Wang, Wenshuo Guo**
University of California, Berkeley
{serenalwang,wsguo}@berkeley.edu

## Abstract

In machine learning applications such as ranking fairness or fairness over intersectional groups, one often encounters optimization problems with extremely large numbers of constraints. In particular, with ranking fairness tasks, there may even be a variable number of constraints, e.g. one for each query in the training set. In these cases, the standard approach of optimizing a Lagrangian while maintaining one Lagrange multiplier per constraint may no longer be practical. Our proposal is to associate a feature vector with each constraint, and to learn a "multiplier model" that maps each such vector to the corresponding Lagrange multiplier. We prove optimality, approximate feasibility and generalization guarantees under assumptions on the flexibility of the multiplier model, and empirically demonstrate that our method is effective on real-world case studies.

## 1 Introduction

Constrained optimization has proven to be useful for a variety of machine learning applications, including churn reduction, Neyman-Pearson classification, and the imposition of statistical fairness constraints [e.g. 1, 2]. In such problems, there are generally only a handful of constraints: for example, in a fairness problem, there will typically be only one constraint per protected group. As a result, while optimizing such a constrained ML problem *is* more difficult than optimizing an unconstrained problem, the difference is usually relatively small.

In some cases, however, it may be desirable to include an extremely large number of constraints. For example, in a fairness problem in which the data are partitioned into protected groups in multiple distinct ways (e.g. race, gender, age buckets, etc.), enforcing a constraint independently for each of the individual groups may not satisfy the constraint for each *intersection* of those groups [3]. Therefore, it may be necessary to impose a separate constraint for each such intersection. As one tries to impose statistical fairness requirements along an increasing number of such dimensions, the total number of constraints can quickly get out of hand.

Despite the difficulties with such a setting, the situation in such an intersectional statistical fairness problem isn't as bad as it *could* be, since there are still a known, finite number of constraints. In other settings, even this might not be the case. In many ranking problems, for example, the workflow is that a "query" is provided, with each query being associated with some set of "documents", which are then handed off to a model to be ranked. Here, we're mainly interested in the problem of imposing fairness constraints on a *per-query* basis. When statistical fairness constraints (such as ranking analogues to demographic parity or equal opportunity [4], adapted to the ranking setting as in Narasimhan et al.

[5]) are imposed on a ranking problem, they are generally enforced *on average* across all queries [e.g. 6, 5]. As a result, any *particular* query might be "unfair" (w.r.t. whatever fairness metric is being used), as long as this unfairness averages out. Existing work on per-query fairness metrics is, so far as we're aware, restricted to post-processing approaches [7–9]. Our goal is to formulate such a problem as an explicit constrained optimization problem, with one (or potentially more) fairness constraints per query. One of the main difficulties of this setting is that, while one could think of there being one constraint per query in the *training set*, it would be more accurate to imagine that there are potentially an infinite number of constraints, one for each *possible* query, and those corresponding to the training set are merely those that we happen to be able to easily observe. This raises the interesting question of how well such constraints *generalize*.

A third example we consider is a variant of *robust optimization*, specifically in a statistical fairness setting in which protected group information isn't available, but we *do* have access to correlated "noisy" features. In one canonical example, "zip codes" serve as a noisy proxy for "race". Wang et al. [10] proposed using a robust-optimization-like approach to this problem, in which the correlations between the "true" and "noisy" groups were assumed to be known (or estimated from a side-dataset), and the fairness constraints were required to hold for the *worst* true-group labeling that was consistent with both the known proxy-group labeling and these known correlations. Applied to their problem, our approach differs only formally, in that instead of imposing one constraint for the *worst* consistent labeling, we have a separate constraint for *each* such labeling, and want *all* of them to hold.

The unifying property of the above tasks is that each includes either a very large constraint set, or worse, an infinite one. In the former case, the model being learned could easily have insufficient capacity to satisfy all of the constraints simultaneously (i.e. the problem could be *infeasible*), while in the latter case, unless the constraints are highly-structured, expecting all of them to hold is unrealistic. For this reason, our approach *does not attempt* to satisfy all of the constraints simultaneously. Instead, it is based on the Lagrangian formulation, and parameterizes the Lagrange multipliers using a *model*. In general, as the complexity of this model increases, so too does its ability to satisfy the constraints. If it is under-parameterized (as will typically be the case), then some constraints are likely to be violated (Section 4 provides some intuition on how our approach copes with this situation). However, if we use e.g. a neural network, then because the same set of weights will be used for every constraint, the model will be capable of learning relationships and redundancies between constraints, and therefore could perform better than its apparent complexity might indicate.

We make five main contributions: (i) in Section 3, we introduce the idea of using the Lagrangian formulation when the Lagrange multipliers are not taken to be a simple vector, but instead are the output of a model, given some set of features; (ii) in Definition 1 of Section 5, we introduce a new notion of how one can *measure* constraint violations in our Lagrangian-model setting, in a way that permits theoretical results to be proved; (iii) later in Section 5, we prove a suboptimality and infeasibility guarantee; (iv) in Section 5.2 we prove a generalization result that applies to the per-query ranking fairness setting described above; (v) in Section 6, we provide an extensive experimental evaluation of several of the highly-constrained settings we have discussed.

## 2   Related Work

There is a significant body of work that handles heavily-constrained problems by projecting onto each constraint (see e.g. Wang and Bertsekas [11, 12] and references therein). The use of a Lagrangian-like formulation is also popular, but we are aware of relatively little work that tries to reduce the complexity of the Lagrange multiplier space itself. A near-exception is Cotter et al. [13], which points out that one can partition the constraint set, and then associate each partition element with the *maximum* over the constraint functions of its components, thereby creating an equivalent problem with only as many constraints as partition elements.

In recent years, with the growing popularity of various statistical fairness metrics, Lagrangian-like approaches have started to be applied to fairness problems, with the goal being to satisfy the constraints as a part of the main training pipeline, instead of being applied as a post processing step [14–17]. In such approaches, intersectional groups have always been a potential problem, as they require constraining an exponential number of group intersections, many of which may contain very few training examples. As a remedy, Kearns et al. [3] propose imposing constraints instead on subgroups defined by a class of membership functions over protected attributes. They then show how

one can provably learn a classifier that satisfies false positive rate and coverage constraints over this class as long as it has bounded VC dimension. They additionally assume access to an oracle to find the maximally violated constraint at each step of their optimization. Hebert-Johnson et al. [18] also propose a similar approach for calibration-based fairness constraints.

Like these prior methods, our approach also seeks to exploit the redundancies in the dual solution, but does so by limiting the flexibility of the Lagrange multiplier model used to enforce constraints. As a result, our optimization strategy is simpler, and does not require keeping track of the most violated constraint. We are thus able to handle a more general set of problems, including the ranking fairness example in which we have one constraint per query, and finding the maximum violation over *all* queries seen so far would be prohibitively expensive. The downside to this generality is that our approach is only able to satisfy the constraints approximately, but as we show in our experiments, the quality of the approximation improves with the capacity of the Lagrange multiplier class used.

Another closely related area of research is fairness in ranking [e.g. 19, 6, 8, 20, 5, 21]. Many existing proposals for ranking fairness impose constraints *on average*, instead of *per query*, including more recent policy gradient based methods [20]. The exceptions that we are aware of that do handle per-query constraints do not impose them during training, but instead later while post-processing a pre-trained scoring model [7–9]. Our approach differs from both of these lines of research in that it attempts to impose per-query fairness constraints *during training*.

## 3   Modeling Lagrange Multipliers

Our goal is to find a parameter vector $\theta \in \Theta$ that minimizes an objective function $g : \Theta \to \mathbb{R}$, subject to a total of $m$ inequality constraints defined by the functions $h_i : \Theta \to \mathbb{R}$ for $i \in [m]$:

$$\underset{\theta \in \Theta}{\text{minimize}} \; g\left(\theta\right) \quad \text{s.t.} \quad \forall i \in [m] . h_i\left(\theta\right) \leq 0 \tag{1}$$

We do not assume that any of these functions are convex. Typically, the objective function $g$ will be an average loss over some dataset, but our main focus is on the constraints. Our interest is in problems for which $m$ is extremely large (or even unknown). For example, there might be one constraint per training example, in which case we could define $h_j\left(\theta\right) = h^\dagger\left(x_j; \theta\right)$ to be a function of the associated feature vector $x_j$. Alternatively, in a ranking problem, we might have one constraint per query, and could take $h_j\left(\theta\right) = h^\dagger\left(z_j; \theta\right)$, where $z_j$ would contain features summarizing the $j$th query.

The usual approach to tackling such problems is to formulate the Lagrangian:

$$\mathcal{L}\left(\theta, \lambda\right) := g\left(\theta\right) + \sum_{i=1}^{m} \lambda_i h_i\left(\theta\right) \tag{2}$$

and to then seek a mixed (due to the non-convexity of $g$ and the $h_i$s, see e.g. Cotter et al. [15]) Nash equilibrium of the game that results from minimizing the Lagrangian in $\theta$ and maximizing it in $\lambda$. If we can find such an equilibrium, then the random variable corresponding to the $\theta$ portion of the mixed equilibrium will be feasible and optimal, in expectation. One way to accomplish this would be to iteratively choose the best-response in $\theta$, and perform a stochastic gradient ascent update in $\lambda$, resulting in a long sequence of $\theta$ and $\lambda$, over which the uniform distribution will be an approximate mixed Nash equilibrium [e.g. 22, 23].

In our setting, $m$ might be extremely large, so this approach can easily fail in practice, since there will simply be far too many Lagrange multipliers. For example, if the training data are provided as a continuous stream of *i.i.d.* samples, and there is one constraint per training example (or per-query, in the ranking setting), then the approach described above would perform *only one* update to each Lagrange multiplier (occurring the first and only time that it is encountered), and the constraints will be all-but ignored.

Instead, we propose modifying the Lagrangian in such a way that $\lambda$ is no longer an element of $\mathbb{R}_+^m$, but is instead represented as a *function* $\vec{\lambda} : \Gamma \to \mathbb{R}_+^m$, where $\Gamma$ is a set of learned parameters. The Lagrangian can then be approximated as:

$$\tilde{\mathcal{L}}\left(\theta, \gamma\right) = g\left(\theta\right) + \left\langle \vec{\lambda}\left(\gamma\right), \vec{h}\left(\theta\right) \right\rangle \tag{3}$$

where, to simplify the notation, we've written the sum of Equation 2 as an inner product, with $\vec{\lambda}(\gamma) = [\lambda_1(\gamma), \lambda_2(\gamma), \ldots, \lambda_m(\gamma)]^T$ and $\vec{h}(\theta) = [h_1(\theta), h_2(\theta), \ldots, h_m(\theta)]^T$. Notice that we recover the usual Lagrangian by defining $\Gamma = \mathbb{R}^m_+$ and taking $\vec{\lambda}$ to be the identity function.

## 4 Simple Example: Linear Multiplier Model

As a thought experiment, imagine that the $i$th constraint is associated with a non-negative feature vector $M_{i,:} \in \mathbb{R}^{\tilde{m}}_+$, where $\tilde{m} \ll m$, and that we've stacked these feature vectors in the matrix $M \in \mathbb{R}^{m \times \tilde{m}}_+$ with one row for each constraint. We'll take our Lagrange multiplier model to have a particularly simple form, with $\gamma \in \Gamma := \mathbb{R}^{\tilde{m}}_+$ being the parameter vector that we will learn, and $\vec{\lambda}$ being a *linear* function of these parameters, i.e. $\vec{\lambda}(\gamma) := M\gamma$. With this Lagrange multiplier model, the Lagrangian becomes:

$$\tilde{\mathcal{L}}(\theta, \gamma) = g(\theta) + \left\langle M\gamma, \vec{h}(\theta) \right\rangle = g(\theta) + \left\langle \gamma, M^T \vec{h}(\theta) \right\rangle.$$

Interestingly, the latter expression is the "usual" Lagrangian of the constrained problem:

$$\underset{\theta \in \Theta}{\text{minimize}}\ g(\theta) \quad \text{s.t.} \quad \forall i \in [\tilde{m}] \cdot \left\langle M_{:,i}, \vec{h}(\theta) \right\rangle \leq 0. \tag{4}$$

Hence, there is an alternative interpretation of a linear Lagrange multiplier model: we can instead consider it to be the *usual* (non-approximated) Lagrangian, with a linear transformation being applied to the *constraint functions*, instead of the $\gamma$s.

**Shared multipliers:** As an even simpler special-case, imagine that $\gamma$ consists of a set of Lagrange multipliers that are *shared* among the constraints, i.e. for each $i \in [\tilde{m}]$, there is a set $S_i \subseteq [m]$ containing the indices of all constraints that will use $\gamma_i$ as their Lagrange multiplier, with the $S_i$s forming a partition of $[m]$. Then $M$ will have exactly one element in each row equal to one and the others equal to zero, and the resulting constrained problem (Equation 4) will be:

$$\underset{\theta \in \Theta}{\text{minimize}}\ g(\theta) \quad \text{s.t.} \quad \forall i \in [\tilde{m}] \cdot \frac{1}{|S_i|} \sum_{j \in S_i} h_j(\theta) \leq 0.$$

Here, we've written the constraints as an average, rather than a sum, since an arbitrary positive scaling factor can be applied to each constraint without changing the constrained problem.

As this example illustrates, this highly-simplified approach of "sharing" Lagrange multipliers has an important property: it *fails gracefully*. Specifically, when the $\Gamma$ space doesn't have enough capacity to satisfy all of the constraints, it falls back on the behavior of satisfying particular *average* constraints.

**Discussion:** Our actual proposal (Section 3) is to associate a feature vector with each constraint, and to learn a not-necessarily-linear model (a neural network, for example) that learns the Lagrange multipliers based on this data. Hence, information could be shared between constraints in a more nuanced manner than a simple average (or a weighted average, as will be the case for a general non-negative $M$). However, we believe, based on the above discussion, that it will still be possible to interpret such a model as a transformation of the *constraints*, and that our proposal will therefore still tend to "fail gracefully". In Section 6.1, we provide experimental evidence that this intuition holds in practice. Making this notion precise is, we believe, an exciting area for future research.

## 5 Algorithm & Analysis

In the previous section, we observed that an under-parameterized Lagrange multiplier model cannot be expected to satisfy all of the constraints simultaneously (and expressed our hope that it would still tend to "fail gracefully"). For this reason, we need to *define* how we will measure the magnitudes of the constraint violations.

**Definition 1.** (**Violation function**) *We measure the violation in the $m$ constraints by a function $\Phi : \mathbb{R}^m \to \mathbb{R}_+$, which for any random variable $\theta^*$ taking values in parameters $\Theta$ satisfies:*

$$\Phi\left(\mathbb{E}_{\theta \sim \theta^*}\left[\vec{h}(\theta)\right]\right) \leq \max_{\gamma \in \Gamma} \left\langle \vec{\lambda}(\gamma), \mathbb{E}_{\theta \sim \theta^*}\left[\vec{h}(\theta)\right] \right\rangle.$$

Table 1: Examples of violation functions that satisfy Definition 1. Here, $\theta^*$ is a random variable taking values on $\Theta$, $\mathbf{1}\{\cdot\}$ is an indicator function, and $(z)_+$ retains the positive entries of vector $z$ and replaces the others with 0. In the fourth row, $p, q \geq 1$ and $1/p + 1/q = 1$. The four examples of $\Phi$ include the positive part of the average constraint violation, the $L_1$-norm of the positive violations, the squared $L_2$-norm of the positive violations, and the $p$-norm violation measured on a linear transformation of the constraints (see linear model example in Section 4).

| Assumption on multiplier model $\lambda(\cdot)$ | $\Phi(z)$ |
|---|---|
| $\Gamma = [0, 1/m]$ and $\forall x, \lambda(\gamma; x) = \gamma$ | $\left(\frac{1}{m} \sum_{j=1}^m z_j\right)_+$ |
| $\forall \theta^*, \exists \gamma$ s.t. $\vec{\lambda}(\gamma) = \mathbf{1}\left\{\mathbb{E}_{\theta \sim \theta^*}\left[\vec{h}(\theta)\right] > 0\right\}$ | $\|(z)_+\|_1$ |
| $\forall \theta^*, \exists \gamma$ s.t. $\vec{\lambda}(\gamma) = \left(\mathbb{E}_{\theta \sim \theta^*}\left[\vec{h}(\theta)\right]\right)_+$ | $\|(z)_+\|_2^2$ |
| $\Gamma = \left\{\gamma \in \mathbb{R}_+^{\tilde{m}} : \|\gamma\|_q \leq 1\right\}$ and $\vec{\lambda}(\gamma) = M\gamma$ for $M \in \mathbb{R}_+^{m \times \tilde{m}}$ | $\|(M^\top z)_+\|_p$ |

One should observe that $\Phi$ is needed purely for our theory: it has no influence on the algorithm, which depends only on the Lagrange multiplier model *itself*. Given the form of this model (i.e. its function class), there are *many* compatible choices of $\Phi$, each of which determines the sort of infeasibility and generalization guarantees that we can achieve. For example, if we choose the Lagrange multiplier models to be constant functions of the form $\lambda(\gamma; x) = \gamma$, for $\gamma \in [0, 1/m]$, we would be constraining the average constraint function $\frac{1}{m} \sum_{j=1}^m h_j(\theta)$ to be non-negative, and a choice of $\Phi$ that satisfies Definition 1 in this case is $\Phi(z) = \left(\frac{1}{m} \sum_{j=1}^m z_j\right)_+$. In Table 1, we give examples of $\Phi$ satisfying Definition 1 for some other simple function classes. For more complicated function classes, the form of $\Phi$ may not be known, but can still be reasoned about formally.

Ideally, we would like $\Phi$ to measure the maximum constraint violation, but this can be a very strong requirement, since it essentially states that the Lagrange multiplier model has sufficient capacity to penalize violations in each of the $m$ constraints, even when $m$ is extremely large. With that said, Table 1 shows that we can come up with reasonable candidates for $\Phi$ for some settings. Notice that, in this table, none of the provided choices of $\lambda$ can grow without bound, which should be a cause for concern, since we might expect it to be necessary for $\lambda$ to be very large for nearly-infeasible problems. For this reason, we actually scale the constraint portion of the Lagrangian by an additional positive scaling factor $(R)$ in the upcoming theorem.

We are now ready to prove our main result, which builds on Agarwal et al. [14]. We will make a mild assumption that the Lagrange multiplier *model* is capable of representing an all-zero $\lambda$:

**Assumption 1.** *There exists a $\gamma \in \Gamma$ such that $\vec{\lambda}(\gamma) = 0$.*

**Theorem 1.** *For any given radius $R > 0$, suppose that $\theta^*$ and $\gamma^*$ are random variables taking values from $\Theta$ and $\Gamma$ (respectively), and that they're an $R\epsilon$-approximate mixed Nash equilibrium, i.e.:*

$$\sup_{\gamma \in \Gamma} \mathbb{E}\left[g(\theta^*) + R\left\langle \vec{\lambda}(\gamma), \vec{h}(\theta^*)\right\rangle\right] - \inf_{\theta \in \Theta} \mathbb{E}\left[g(\theta) + R\left\langle \vec{\lambda}(\gamma^*), \vec{h}(\theta)\right\rangle\right] \leq R\epsilon. \quad (5)$$

*Notice that an additional weight of $R$ has been applied to the constraint portion of the Lagrangian. Assuming that the problem is feasible and that Assumption 1 holds, and choosing $R = 1/\sqrt{\epsilon}$, we have that $\theta^*$ will be approximately optimal, in expectation:*

$$\mathbb{E}[g(\theta^*)] \leq \inf_{\theta \in \Theta : \forall i \in [m], h_i(\theta) \leq 0} g(\theta) + \sqrt{\epsilon}.$$

*Further for any violation function $\Phi : \mathbb{R}^m \to \mathbb{R}_+$ that satisfies Definition 1, we have that $\theta^*$ will likewise be approximately feasible (measured in terms of $\Phi$):*

$$\Phi\left(\mathbb{E}\left[\vec{h}(\theta^*)\right]\right) \leq G\sqrt{\epsilon} + \epsilon.$$

*Where $G \geq \sup_{\theta \in \Theta} g(\theta) - \inf_{\theta \in \Theta} g(\theta)$ bounds the range of the objective function g.*

*Proof.* In Appendix A. $\qquad \square$

This theorem shows that, assuming that Assumption 1 holds and a suitable $\Phi$ exists, an approximate Nash equilibrium will be approximately feasible and optimal. Other refinements from the literature, such as the use of a non-zero-sum formulation with proxy-constraints [15, 17], or to improve constraint generalization [16], could be integrated with some extra effort.

## 5.1 Algorithm

Theorem 1 does not describe how one can *find* an approximate mixed Nash equilibrium, particularly given that the objective and constraint functions are not assumed to be convex. This however, is well-studied: as we mentioned in Section 3, it's straightforward to see that if we treat the Lagrangian as a two-player game, and have the players iteratively play against each other, with the $\theta$ player using best-response (i.e. choosing the global minimum in $\theta$ for a fixed $\gamma$—this is necessary due to the non-convexity of the objective $g$ and constraints $h_i$), and the $\gamma$ player playing e.g. stochastic gradient ascent (assuming that the Lagrange multiplier model $\lambda$ is a concave function of $\gamma$), then the uniform distribution over the resulting sequence of iterates will define a pair of random variables $\theta^*$ and $\gamma^*$ that form an approximate mixed Nash equilibrium, and, by Theorem 1, $\theta^*$ will be approximately feasible and optimal. In practice, having both players use e.g. SGD, and ignoring the non-convexity, often works well [e.g. 24], and this is the approach that we take in the experiments of Section 6.

We seek a mixed equilibrium because our Lagrangian is non-convex in $\theta$, so a pure Nash equilibrium might not even *exist* [15]. A mixed equilibrium is therefore difficult to avoid, but it does come with the unfortunate consequence that it results in a *stochastic* classifier: instead of finding a single parameter vector $\theta$, we find a *distribution* $\theta^*$ over multiple parameter vectors. At evaluation time, each time that we receive a new example, we will sample a $\theta \sim \theta^*$, and then classify the example using the sampled $\theta$. Deterministic classifiers are of course much more convenient, and we've found that ignoring this issue by deterministically taking the last iterate (instead of the uniform distribution over all iterates), often (but not always) works well. Alternatively, one could first create a stochastic classifier by finding a mixed equilibrium, and then convert it into a deterministic classifier using e.g. the procedure of Cotter et al. [25].

We include further details in Appendix C.

## 5.2 Generalization

In heavily constrained problems, one obvious area of concern is *generalization*. The naïve approach would be to determine a separate generalization bound for each constraint, and to then apply the union bound to find a common bound on *all* constraints. However, this might not scale well to heavily-constrained problems like those that we discussed in Section 1, and in particular, cannot be applied *at all* in the setting in which we have (implicitly) an infinite number of constraints, e.g. when there is one constraint per example (or per query).

In Appendix B, we consider the setting in which the number of constraints is constant (but large), and each constraint function is represented as an *expectation* over the data distribution (or a subset of it, as in e.g. fairness over intersectional groups). Many previous papers describe how such constraints can be constructed [e.g. 1, 2, 24]. Interestingly, it turns out that it is possible to bound the generalization of constraints in such a way that the bound depends not on the number of constraints, but rather on a complexity measure of the function class associated with the Lagrange multiplier model, although the performance is measured indirectly, in terms of a particular choice of $\Phi$.

Below, we will study the constraint generalization of our approach assuming that there is one constraint per example (or per query), and that the constraint function itself only depends on the particular example (or query) with which it is associated (e.g. ranking fairness). In the experiments of Section 6.3, we construct constraints in the ranking setting using an approach that is basically identical to that of Narasimhan et al. [5], except that we will include one constraint per query, instead of averaging over the entire dataset. One interesting thing about this setting is that, while during training we will only have access to those constraints that correspond to queries that occur in the training set, implicitly there is a constraint for every *possible* query. Since, at evaluation time, we will observe queries, and therefore constraints, that were not present in the training dataset, it's very important to ensure that the constraints *generalize well*.

We will use $\mathcal{D}$ to denote the underlying data distribution over instances $x \in \mathcal{X}$. We would like to find parameters $\theta$ that satisfy the following constraints: for each instance $x \in \mathcal{X}$, $h(\theta; x) \leq 0$. For the constraint associated with each $x$, we will use the Lagrange multiplier $\lambda(\gamma; x)$. Similar to Definition 1, we will measure the overall constraint violations with a particular choice of $\Phi$:

$$\Phi(\theta) = \max_{\gamma \in \Gamma} \mathbb{E}_{x \sim \mathcal{D}} \left[ \lambda(\gamma; x) h(\theta; x) \right].$$

Notice that unlike Definition 1, we've written $\Phi$ in terms of $\theta$ instead of the constraint functions $h(\theta; x)$, but the $h$s are themselves functions of $\theta$, so this is nothing but a change of notation. In practice, we are given a sample $S$ of $n$ examples drawn from $\mathcal{D}$, and we instead work with the empirical constraint violation, given by:

$$\hat{\Phi}(\theta) = \max_{\gamma \in \Gamma} \frac{1}{n} \sum_{j=1}^{n} \lambda(\gamma; x_j) h(\theta; x_j).$$

We now bound the gap between the expected and empirical constraint violations.

**Theorem 2.** *Suppose* $\sup_{\gamma, x} \lambda(\gamma; x) \leq B_\lambda$ *and* $\sup_{\theta, x} |h(\theta; x)| \leq B_\theta$. *With probability at least* $1 - \delta$ *over the* i.i.d. *draw of* $S \sim \mathcal{D}^n$, *for all* $\theta \in \Theta$:

$$\Phi(\theta) \leq \hat{\Phi}(\theta) + \widehat{\mathcal{R}}_n(\mathcal{H}_\Theta) + \widehat{\mathcal{R}}_n(\Lambda_\Gamma) + B_\lambda B_\theta \sqrt{\frac{\log(1/\delta)}{n}},$$

*where* $\widehat{\mathcal{R}}_n(\mathcal{H}_\Theta)$ *is the empirical Rademacher complexity of the class of constraint functions* $\{x \mapsto h(\theta; x) | \theta \in \Theta\}$ *for any sample of size* $n$ *and* $\widehat{\mathcal{R}}_n(\Lambda_\Gamma)$ *is the empirical Rademacher complexity of the class of multipliers* $\{x \mapsto \lambda(\gamma; x) | \gamma \in \Gamma\}$.

*Proof.* In Appendix A. ☐

## 6 Experiments

We present experiments on (i) fairness task with intersectional protected groups, (ii) a fairness task with noisy protected groups, and (iii) a ranking fairness task with per-query constraints. We report running times and additional experimental details in Appendices D–F.[1]

### 6.1 Fairness Constraints on Intersectional Groups

We first demonstrate how our approach *fails gracefully* as we decrease the complexity of the Lagrangian model. For this, we consider the task of training a linear classifier to predict crime rate for a community, subject to intersectional group fairness constraints. We use the Communities and Crime dataset [26], which contains 1,994 communities in the US described by 140 features, and seek to predict the per capita crime rate for each community. As with prior work [e.g. 15], we label the communities with a crime rate above the 70th percentile as 'high crime' and the others as 'low crime'. We form intersectional groups based on the percentages of the Black, Hispanic and Asian populations $(z_1, z_2, z_3)$ in a community. We define a protected group using three thresholds $(t_1, t_2, t_3)$, and include in it all communities where the race percentages $z_k \geq t_k, \forall k \in [3]$. We consider a total of 1000 threshold combinations $(t_1, t_2, t_3)$ from $[0, 1]^3$, refer to each such group by $G_{t_1, t_2, t_3}$, and impose a constraint on its error rate: $\text{error}(G_{t_1, t_2, t_3})$. Specifically, we seek to minimize the overall error rate, subject to the constraint $\text{error}(G_{t_1, t_2, t_3}) \leq \text{error}(\text{ALL}) + 0.01$, for all $(t_1, t_2, t_3)$, except those groups that contain less than 1% of the data. We end up with a total of 535 groups.

We solve this constrained optimization problem by using a multiplier model that takes the three thresholds $(t_1, t_2, t_3)$ as input and assigns a Lagrange multiplier to the corresponding error rate constraint. During training, we sample a threshold combination at random and perform gradient updates on the Lagrangian for the sampled constraint. We experiment with five multiplier architectures, ranging from under-parameterized to over-parameterized models. This includes a *common* multiplier for all constraints, a linear model, and neural networks with one, two and three hidden layers (with 50 nodes each). We compare with an *unconstrained* baseline that optimizes error rate without constraints.

The violation in the constraint for group $G_{t_1, t_2, t_3}$ is measured as $\text{error}(G_{t_1, t_2, t_3}) - \text{error}(\text{ALL}) - 0.01$, where a negative value indicates that the constraint is satisfied. The plots in Figure 1 shows different percentiles of the constraint violations, with the left-most point showing the median violation and the right-most point showing the 95th percentile violation. The unconstrained classifier incurs positive violations for all percentiles, while the 3-hidden layer multiplier model is able to satisfy most constraints on the training set. Interestingly, as we move from a 3-hidden layer model to a common multiplier, the higher percentile violations go up, but the lower percentile violations are still

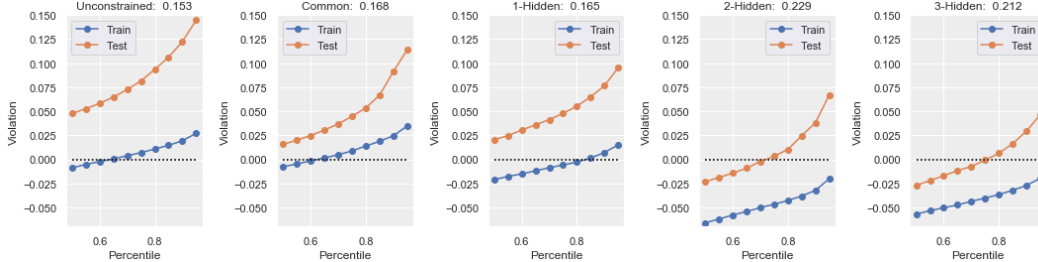

Figure 1: Plots of $p$th percentile violations as a function of $p$ (averaged over 5 trials) for the intersectional group constraints on the Communities & Crime dataset. The overall test error rates (*lower* better) are reported in the titles. Negative violations indicate the constraints are satisfied.

Table 2: Comparison with the FairFictitiousPlay approach of Kearns et al. [3] with a brute-force oracle to find the most-violated constraint on the Communities & Crime intersectional fairness task. We report the error rates and 95th percentile violations (means and standard errors over 5 trials).

| Method | Train | | Test | |
| --- | --- | --- | --- | --- |
| | Error | 95th Perc. Violation | Error | 95th Perc. Violation |
| Unconstrained | **0.11 ± 0.002** | 0.05 ± 0.006 | **0.15 ± 0.003** | 0.09 ± 0.012 |
| Common multiplier | 0.14 ± 0.002 | 0.03 ± 0.003 | 0.17 ± 0.006 | 0.11 ± 0.015 |
| Linear | 0.14 ± 0.003 | 0.03 ± 0.004 | 0.15 ± 0.005 | 0.14 ± 0.004 |
| 1-hidden-layer | 0.14 ± 0.013 | 0.02 ± 0.005 | 0.16 ± 0.010 | 0.10 ± 0.015 |
| 2-hidden-layer | 0.21 ± 0.013 | -0.02 ± 0.006 | 0.23 ± 0.012 | 0.07 ± 0.021 |
| 3-hidden-layer | 0.20 ± 0.008 | **-0.02 ± 0.004** | 0.21 ± 0.009 | 0.05 ± 0.017 |
| FairFictitiousPlay | 0.20 ± 0.010 | **-0.02 ± 0.004** | 0.22 ± 0.011 | **0.04 ± 0.008** |

better than the unconstrained approach. Thus with a simpler architecture, the Lagrange multiplier model tries to satisfy a simpler transformation of the constraints, and is better-able to fail gracefully (Section 4). Despite the large number of constraints, the overall error doesn't reach the trivial error rate (0.3 for an all-negatives classifier), likely because of redundancies in the constraints.

We also compare our results with the FairFictitiousPlay algorithm of Kearns et al. [3], who also impose constraints on intersectional groups, but assume access to an oracle for finding the maximally violated constraint at each step. We implement this oracle using a brute-force search. As seen in Table 2, despite not having access to a brute-force oracle, the 3-hidden layer multiplier model is able to satisfy the constraints almost as well as FairFictitiousPlay, while yielding a similar error rate.

## 6.2 Fairness Constraints with Noisy Group Memberships

We next apply our proposed approach to impose group-based fairness constraints when the protected groups are noisy. The goal is to equalize true positive rates over the unknown, "true" group membership assignments, when we only have access to known noisy group membership assignments. Wang et al. [10] propose robust-optimization-like approaches for this problem. We replicate their setup, but propose a different optimization strategy. Denoting the true group for an instance by $G \in [k]$ and the noisy group by $\hat{G} \in [k]$, we assume access to the marginal probabilities $P(G = i | \hat{G} = j), \forall i, j \in [k]$. Then given a dataset $\{(x_i, y_i, \hat{G}_i)\}_{i=1}^n$ with noisy group memberships, we consider the set of all candidate true group memberships $\mathcal{Z} = \left\{ \{G_i^{(1)}\}_{i=1}^n, \{G_i^{(2)}\}_{i=1}^n, \dots \right\}$ that satisfy the marginal probabilities, and seek to impose the fairness constraints on each of these candidate memberships. Specifically, we wish to solve the following optimization problem:

$$\underset{\theta \in \Theta}{\text{minimize}}\ g(\theta) \quad \text{s.t.} \quad \forall Z \in \mathcal{Z},\ j \in [k].\ h_j(\theta, Z) \le 0, \tag{6}$$

where $h_j(\theta, Z)$ denotes an equal opportunity fairness constraint evaluated on group $j$ with the candidate group memberships $Z$. Note that the number of constraints can be exponential in the number of instances $n$, and hence its impractical to maintain one Lagrange multiplier per constraint. Instead, we use a model that maps a group membership assignment $Z$ to a Lagrange multiplier, and we sample group memberships from $\mathcal{Z}$ at random to perform gradient updates.

We use the UCI Adult dataset [26]. We optimize for the error rate subject to *equality of opportunity* [4] on three race groups 'white', 'black' and 'other', and train a linear classification model. We use

Table 3: Equal opportunity constraints on Adult with different group noise levels (mean and standard error over 10 trials). We report the test error rates (*lower* better) and test equal opportunity violations evaluated with the *true* groups. A negative fairness violation indicates the constraints are satisfied.

| | DRO [10] | | SoftAssign [10] | | Proposed | |
|---|---|---|---|---|---|---|
| Noise | Error rate | Violation | Error rate | Violation | Error rate | Violation |
| 0.1 | $0.152 \pm 0.001$ | $0.002 \pm 0.019$ | $\mathbf{0.148 \pm 0.001}$ | $-0.048 \pm 0.002$ | $\mathbf{0.147 \pm 0.002}$ | $0.015 \pm 0.035$ |
| 0.2 | $0.200 \pm 0.002$ | $-0.045 \pm 0.003$ | $\mathbf{0.157 \pm 0.003}$ | $-0.048 \pm 0.002$ | $0.168 \pm 0.006$ | $-0.004 \pm 0.013$ |
| 0.3 | $0.216 \pm 0.010$ | $-0.044 \pm 0.004$ | $\mathbf{0.158 \pm 0.005}$ | $0.002 \pm 0.030$ | $0.165 \pm 0.003$ | $-0.016 \pm 0.002$ |
| 0.4 | $0.209 \pm 0.006$ | $-0.019 \pm 0.031$ | $0.188 \pm 0.003$ | $-0.016 \pm 0.016$ | $\mathbf{0.157 \pm 0.002}$ | $-0.005 \pm 0.028$ |
| 0.5 | $0.219 \pm 0.012$ | $-0.030 \pm 0.032$ | $0.218 \pm 0.002$ | $0.004 \pm 0.006$ | $\mathbf{0.189 \pm 0.006}$ | $-0.020 \pm 0.009$ |

Table 4: Pairwise ranking fairness with per-query constraints on MSLR-WEB10K (averaged over 5 trials). Constraint violation is measured as $|\text{err}_{0,1} - \text{err}_{1,0}| - 0.25$ (*lower* is better).

| | Pairwise Error | | 90th Percentile Violation | |
|---|---|---|---|---|
| Method | Train | Test | Train | Test |
| Unconstrained | $\mathbf{0.386 \pm 0.010}$ | $\mathbf{0.394 \pm 0.008}$ | $0.068 \pm 0.003$ | $0.282 \pm 0.029$ |
| Narasimhan et al. [5] | $0.396 \pm 0.014$ | $0.400 \pm 0.014$ | $0.060 \pm 0.004$ | $0.246 \pm 0.025$ |
| Proposed | $0.406 \pm 0.012$ | $0.412 \pm 0.011$ | $\mathbf{0.040 \pm 0.005}$ | $\mathbf{0.195 \pm 0.023}$ |

a linear multiplier model which takes as input an $n$-dimensional boolean encoding of the sampled group membership assignment $Z$. We compare our approach with the two approaches proposed by Wang et al. [10], based on distributionally robust optimization (DRO) and "soft" group assignments.

Table 3 shows the error rates, and fairness violations on the true groups $G$ for the trained classifiers, evaluated on the test set. As done by Wang et al. [10], we run five different experiments with five different noise levels added to the true groups in the training set: the "noisy" groups that we observe are a perturbation of the true groups using the given noise level. Higher noise yields a larger candidate set $\mathcal{Z}$. The baseline unconstrained model achieves an error rate of $0.145 \pm 0.004$ and a maximum constraint violation of $0.02 \pm 0.05$ [10]. Most importantly, the proposed approach is able to satisfy the fairness constraints on the true groups not seen during training, albeit with a higher variance for lower noise levels. Table 3 shows that when the noise level is low, the soft group assignments approach from Wang et al. [10] has better performance; when the noise level is high, the proposed approach achieves lower error rates than the more conservative soft assignments approach.

### 6.3 Ranking Fairness with Per-query Constraints

We finally consider the problem of imposing per-query constraints for ranking fairness. We adopt the pairwise ranking fairness setup from Narasimhan et al. [5], but unlike them, we wish to satisfy the fairness constraints per-query, and not on average across all queries. We use the Microsoft Learning to Rank Dataset (MSLR-WEB10K) [27] which contains 136 document features for each query-document pair. Similar to Yadav et al. [20], we divide all documents into two protected groups $G \in \{0, 1\}$ based on the 40th percentile of the QualityScore feature, and binarize the relevance labels. We also consider a second W3C experts ranking dataset [21], which we include in Appendix F.

Denoting the set of document features by $\mathcal{X}$ and query features by $\mathcal{Q}$, the goal is to learn a ranking function $f : \mathcal{X} \times \mathcal{Q} \to \mathbb{R}$ that assigns a score $f(X, Q)$ to a given query-document pair $(X, Q)$. We adopt the pairwise equal opportunity fairness criteria from Narasimhan et al. [5] and measure the group-specific pairwise errors for groups $i, j \in \{0, 1\}$ and a given query $q \in \mathcal{Q}$ as:

$$\text{err}_{i,j}(q) = \mathbb{E}[\mathbf{1}\{f(X, q) < f(X', q)\} | Y > Y', G = i, G' = j, Q = q]$$

where $(X, Y, G)$ and $(X', Y', G')$ are tuples of document features, labels and groups sampled for query $Q = q$. We seek to then optimize the overall pairwise error subject to the constraint that the cross-group errors are similar for each query: $|\text{err}_{0,1}(q) - \text{err}_{1,0}(q)| \leq 0.25$, for all $q \in \mathcal{Q}$.

We use a one-hidden layer multiplier neural network model containing 128 nodes to assign a Lagrange multiplier for each query $q \in \mathcal{Q}$, with the average of the feature vectors within a query used as input. The ranking model is also a one-hidden layer neural network. Table 4 shows the pairwise error rates and 90th percentile query level constraint violations for our proposed method on a training set of 1000 queries and a test set of 100 queries, and compares against unconstrained training and the approach of Narasimhan et al. [5], which enforces the same constraints but *on average*. The proposed method achieves the least 90th percentile violation at the cost of a larger error rate.

## Broader Impact

From an ethical standpoint, we believe that the main contribution of our work is that it will make it easier to approximately impose fine-grained statistical fairness constraints, particularly in the intersectional and ranking settings. The greatest weakness is that we cannot guarantee that every constraint will hold: instead, constraint violations are measured in terms of a function $\Phi$, and while this function could technically be e.g. the magnitude of the most violated constraint, it is unlikely that any practical Lagrange multiplier model will work with such a $\Phi$ when the number of constraints is extremely large. However, as we show in our experiments, for many real-world problems our approach is effective in optimizing for reasonable choices of $\Phi$ (such as the 90-th percentile violation).

A particular advantage of our approach is that we hope that it will *fail gracefully*, i.e. even when all of the constraints cannot be satisfied simultaneously, the observed behavior will still fall-back to something that most would consider "sensible". However, except for the simple types of Lagrange multiplier models considered in Section 4, this remains largely an *intuition*, and while we provide experimental evidence that this intuition holds in practice, making this notion more precise is an exciting area for future research.

## Footnotes

[1]Code available at: https://github.com/google-research/google-research/tree/master/many_constraints

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
