[Supplementary Material]

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

.* We'll mostly follow the analysis of Agarwal et al. [14]. We'll begin by proving the optimality portion. Let $\hat{\theta}$ minimize $g\left(\theta\right)$ over all *feasible* $\theta \in \Theta$ (i.e. all $\theta$s for which $\vec{h}\left(\theta\right) \preceq 0$). Plugging this into the right-hand side of Equation 5, and using Assumption 1:

$$\mathbb{E}\left[g\left(\theta^*\right)\right] \leq g\left(\hat{\theta}\right) + R\epsilon$$

which completes the optimality portion of the proof.

We'll now move on to feasibility. Again plugging in $\hat{\theta}$ on the right-hand side of Equation 5:

$$\sup_{\gamma \in \Gamma} \mathbb{E}\left[g\left(\theta^*\right) + R\left\langle \vec{\lambda}\left(\gamma\right), \vec{h}\left(\theta^*\right)\right\rangle\right] \leq g(\hat{\theta}) + R\epsilon$$

$$\mathbb{E}\left[g\left(\theta^*\right)\right] + R\sup_{\gamma \in \Gamma} \mathbb{E}\left[\left\langle \vec{\lambda}\left(\gamma\right), \vec{h}\left(\theta^*\right)\right\rangle\right] \leq g(\hat{\theta}) + R\epsilon$$

$$R\sup_{\gamma \in \Gamma} \mathbb{E}\left[\left\langle \vec{\lambda}\left(\gamma\right), \vec{h}\left(\theta^*\right)\right\rangle\right] \leq g(\hat{\theta}) - \mathbb{E}\left[g\left(\theta^*\right)\right] + R\epsilon$$

$$\sup_{\gamma \in \Gamma} \mathbb{E}\left[\left\langle \vec{\lambda}\left(\gamma\right), \vec{h}\left(\theta^*\right)\right\rangle\right] \leq G/R + \epsilon.$$

where we used the upper bound on $g$. Further, using Definition 1, we get:

$$\Phi\left(\mathbb{E}\left[\vec{h}\left(\theta^*\right)\right]\right) \leq G/R + \epsilon.$$

Setting $R = 1/\sqrt{\epsilon}$ completes the feasibility portion of the proof. $\qquad\square$

**Theorem 2.** *Suppose $\sup_{\gamma,x} \lambda(\gamma; x) \leq B_\lambda$ and $\sup_{\theta,x} |h(\theta; x)| \leq B_\theta$. With probability at least $1 - \delta$ over the i.i.d. draw of $S \sim \mathcal{D}^n$, for all $\theta \in \Theta$:*

$$\Phi(\theta) \leq \widehat{\Phi}(\theta) + \widehat{\mathcal{R}}_n(\mathcal{H}_\Theta) + \widehat{\mathcal{R}}_n(\Lambda_\Gamma) + B_\lambda B_\theta \sqrt{\frac{\log(1/\delta)}{n}},$$

*where $\widehat{\mathcal{R}}_n(\mathcal{H}_\Theta)$ is the empirical Rademacher complexity of the class of constraint functions $\{x \mapsto h(\theta; x) | \theta \in \Theta\}$ for any sample of size $n$ and $\widehat{\mathcal{R}}_n(\Lambda_\Gamma)$ is the empirical Rademacher complexity of the class of multipliers $\{x \mapsto \lambda(\gamma; x) | \gamma \in \Gamma\}$.*

*Proof.* We would like to bound:

$$\max_{\theta \in \Theta}\left(\Phi(\theta) - \widehat{\Phi}(\theta)\right) \leq \max_{\theta \in \Theta, \gamma \in \Gamma}\left(\mathbb{E}_x\left[\lambda(\gamma; x) h(\theta; x)\right] - \frac{1}{n}\sum_{j=1}^{n} \lambda(\gamma; x_j) h(\theta; x_j)\right). \tag{7}$$

All we need to do is to bound the generalization error for the product $\lambda(\gamma; \cdot)h(\theta; \cdot)$ over $\Theta \times \Gamma$. We first directly apply a result from DeSalvo et al. [28] (Lemma 2) to bound the Rademacher complexity of this product class by sum of the Rademacher complexities of the individual classes. This result states that, if $\mathcal{C} = \{x \mapsto \lambda(\gamma; x)h(\theta; x) | \gamma \in \Gamma, \theta \in \Theta\}$, then the empirical Rademacher complexity of $\mathcal{C}$ for any sample of size $n$ is given by $\widehat{\mathcal{R}}_n(\mathcal{C}) \leq \widehat{\mathcal{R}}_n(\Lambda_\Gamma) + \widehat{\mathcal{R}}_n(\mathcal{H}_\Theta)$.

Equipped with this above result, we can apply standard uniform convergence based techniques to derive a generalization bound for the product $\lambda(\gamma; x)h(\theta; x)$. Using the fact that $\lambda(\gamma; x)h(\theta; x) \leq B_\lambda B_\theta$ for all $\theta$ and $\gamma$, we have with probability at least $1 - \delta$ that, for all $\theta \in \Theta$ and $\gamma \in \Gamma$:

$$\mathbb{E}_x \left[ \lambda(\gamma; x)h(\theta; x) \right] \leq \frac{1}{n} \sum_{j=1}^{n} \lambda(\gamma; x_j)h(\theta; x_j) + \widehat{\mathcal{R}}_n(\Lambda_\Gamma) + \widehat{\mathcal{R}}_n(\mathcal{H}_\Theta) + B_\lambda B_\theta \sqrt{\frac{\log(1/\delta)}{n}}.$$

Plugging this back into Equation 7 completes the proof. $\qquad\square$

# B    Generalization for a Fixed Number of in-Expectation Constraints

In Section 5.2, we considered the one-constraint-per-example (or per-query) setting. Here, we consider the other major setting of interest, in which there are a known, fixed number of queries, and each constraint function is represented as an *expectation* over the data distribution (or a subset of it, as in e.g. fairness over intersectional groups). Examples of how such constraints might be constructed can be found in e.g. Goh et al. [1], Narasimhan [2], Cotter et al. [24].

We would like to find parameters $\theta$ that satisfy a constant number of $m$ constraints, where the $i$th constraint is defined by:

$$h_i(\theta) = \mathbb{E}_{x \sim \mathcal{D}} \left[ \ell_i(x; \theta) \right],$$

where $\ell_i : \mathcal{X} \times \theta \to \mathbb{R}$. In practice, we are given a sample $S = \{x_1, \ldots, x_n\} \subseteq \mathcal{X}$ drawn *i.i.d.* from $\mathcal{D}$, and we find a $\theta$ that satisfies the following empirical constraints instead:

$$h_i(\theta; S) = \frac{1}{n} \sum_{j=1}^{n} \ell_i(x_j; \theta).$$

We would like to bound the violations in the expected constraints in terms of the violations in the empirical constraints. To this end, we measure the expected constraint violation by:

$$\Phi(\theta) = \max_{\gamma \in \Gamma} \frac{1}{m} \left\langle \vec{\lambda}(\gamma), \vec{h}(\theta) \right\rangle$$

and the empirical constraint violation by:

$$\widehat{\Phi}(\theta) = \max_{\gamma \in \Gamma} \frac{1}{m} \left\langle \vec{\lambda}(\gamma), \vec{h}(\theta; S) \right\rangle.$$

We measure the complexity of the Lagrange multiplier model class using its pseudo-dimension.

**Definition 2** (Pseudo-dimension of multiplier class). *We define the pseudo-dimension of the multiplier class $\Gamma$ as the VC-dimension of the function class $\{(i, r) \mapsto \text{sign}(\lambda_i(\gamma) - r) : \gamma \in \Gamma\}$.*

We now present our generalization bound for the constraint violation that is *independent of the number of constraints* and instead depends on the pseudo-dimension of the multiplier class:

**Theorem 3.** *Suppose $\left\| \vec{h}(\theta) - \vec{h}(\theta; S) \right\|_\infty \leq G$ for all $\theta \in \Theta$ and $\left\| \vec{\lambda}(\gamma) \right\|_\infty \leq B$ for all $\gamma \in \Gamma$. Suppose that, for any $\gamma \in \Gamma$, with probability at least $1 - \delta$ over draw of $S \sim \mathcal{D}^n$, for all $\theta \in \Theta$:*

$$\frac{1}{m} \left\langle \vec{\lambda}(\gamma), \vec{h}(\theta) - \vec{h}(\theta; S) \right\rangle \leq \text{GenBnd}(\delta) \tag{8}$$

*where $\text{GenBnd}(\delta)$ is an upper bound on the generalization error of the inner product on the LHS and a decreasing function of $\delta$, and holds for any* particular $\gamma$. *We show that this generalization bound can be "lifted" to one that holds uniformly, with an additional term depending on the complexity of the multiplier model. To this end, fix $\rho > 0$. Then with probability at least $1 - \delta$ over draw of $S \sim \mathcal{D}^n$, for all $\theta \in \Theta$:*

$$\Phi(\theta) \leq \widehat{\Phi}(\theta) + \text{GenBnd}\left( \delta \left( \frac{\rho}{B} \right)^D \right) + G\rho,$$

*where $D$ is the pseudo-dimension of the multiplier class, i.e. the class of functions $\{i \mapsto \lambda_i(\gamma) : \gamma \in \Gamma\}$. Also observe that for most generalization bounds, $\mathrm{GenBnd}(\delta)$ will have a $\delta$-dependence of the form $\log(1/\delta)$.*

*Proof.* We would like to bound:

$$\Phi(\theta) \leq \widehat{\Phi}(\theta) + \max_{\gamma \in \Gamma} \frac{1}{m} \left\langle \vec{\lambda}(\gamma), \vec{h}(\theta) - \vec{h}(\theta; S) \right\rangle. \tag{9}$$

Let $\tilde{\Gamma} \subseteq \Gamma$ be a finite set of "covering centers" such that, for all $\gamma \in \Gamma$ there exists a $\tilde{\gamma} \in \tilde{\Gamma}$ for which $\frac{1}{m} \left\| \vec{\lambda}(\gamma) - \vec{\lambda}(\tilde{\gamma}) \right\|_1 \leq \rho$. Then:

$$
\begin{aligned}
\frac{1}{m} \left\langle \vec{\lambda}(\gamma), \vec{h}(\theta) - \vec{h}(\theta; S) \right\rangle \leq & \frac{1}{m} \left| \left\langle \vec{\lambda}(\tilde{\gamma}), \vec{h}(\theta) - \vec{h}(\theta; S) \right\rangle \right| \\
& + \frac{1}{m} \left\| \vec{\lambda}(\gamma) - \vec{\lambda}(\tilde{\gamma}) \right\|_1 \left\| \vec{h}(\theta) - \vec{h}(\theta; S) \right\|_\infty \\
\leq & \frac{1}{m} \left\langle \vec{\lambda}(\tilde{\gamma}), \vec{h}(\theta) - \vec{h}(\theta; S) \right\rangle + G\rho.
\end{aligned}
$$

Using the above relationship we can approximate the the 'max' over $\Gamma$ in Equation 9 with a 'max' over the covering centers $\tilde{\Gamma}$:

$$\Phi(\theta) \leq \widehat{\Phi}(\theta) + \max_{\gamma \in \tilde{\Gamma}} \frac{1}{m} \left\langle \vec{\lambda}(\gamma), \vec{h}(\theta) - \vec{h}(\theta; S) \right\rangle + G\rho.$$

We then bound the 'max' over the covering centers using the generalization bound in Equation 8 and a union bound over $\tilde{\Gamma}$, and have with probability at least $1 - \delta$:

$$\Phi(\theta) \leq \widehat{\Phi}(\theta) + \mathrm{GenBnd}\left(\frac{\delta}{|\tilde{\gamma}|}\right) + G\rho. \tag{10}$$

A standard bound [e.g. 29] on the size of the covering centers for a function class in terms of its pseudo-dimension $D$ is:

$$\left| \tilde{\Gamma} \right| \leq \mathcal{O}\left( \left(\frac{B}{\rho}\right)^D \right).$$

Substituting this into Equation 10, combined with the observation that $\mathrm{GenBnd}(\cdot)$ will be a decreasing function of its argument, completes the proof. □

## C   Outline of Algorithm

**"Theoretical" algorithm:** We'll begin by showing that the best-response based algorithm that we mentioned in Section 5.1 converges to a mixed Nash equilibrium. Algorithm 1 has the $\theta$-player using best response at each iteration (i.e. finding a global minimizer of the modeled Lagrangian of Equation 3), and the $\gamma$-player using (stochastic) gradient ascent, under the assumption that $\vec{\lambda}$ is a concave function of $\gamma$. The particular choice of algorithm for the $\gamma$-player doesn't matter—we only need it to satisfy a low-regret guarantee:

$$\sup_\gamma \frac{1}{T} \sum_{t=1}^{T} \tilde{\mathcal{L}}(\theta^{(t)}, \gamma) - \frac{1}{T} \sum_{t=1}^{T} \tilde{\mathcal{L}}(\theta^{(t)}, \gamma^{(t)}) \leq \epsilon \tag{11}$$

where $\epsilon$ depends the regret bound of whatever algorithm is being used (for stochastic gradient ascent on a concave objective, it will be $\mathcal{O}(1/\sqrt{T})$). For the $\theta$-player, best-response trivially gives us a no-regret guarantee:

$$
\begin{aligned}
\frac{1}{T} \sum_{t=1}^{T} \tilde{\mathcal{L}}(\theta^{(t)}, \gamma^{(t)}) - \frac{1}{T} \sum_{t=1}^{T} \inf_{\theta \in \Theta} \tilde{\mathcal{L}}(\theta, \gamma^{(t)}) =& 0 \\
\frac{1}{T} \sum_{t=1}^{T} \tilde{\mathcal{L}}(\theta^{(t)}, \gamma^{(t)}) - \inf_{\theta \in \Theta} \frac{1}{T} \sum_{t=1}^{T} \tilde{\mathcal{L}}(\theta, \gamma^{(t)}) \leq& 0
\end{aligned}
\tag{12}
$$

**Algorithm 1** finds an approximate mixed Nash equilibrium of the modeled Lagrangian of Equation 3: $\tilde{\mathcal{L}}(\theta, \gamma) = g(\theta) + \left\langle \vec{\lambda}(\gamma), \vec{h}(\theta) \right\rangle$. As we discussed in Section 5.1, the $\theta$-player plays best response, while the $\gamma$-player plays (stochastic) gradient ascent. The use of best response is necessary because we do not assume that the objective or constraint functions are convex. There is nothing special about gradient ascent, however. In Appendix C, we discuss this further, and include an outline of a convergence proof.

---

$\theta^{(0)} \leftarrow 0$    // or some other initial value ....
$\gamma^{(0)} \leftarrow 0$    // ....
**for** $t = 1$ **to** $T$ **do**
    $J_\gamma \leftarrow$ the Jacobian of $\vec{\lambda}(\gamma^{(t-1)})$     // or a *stochastic* Jacobian
    $\Delta_\gamma \leftarrow J_\gamma^T \vec{h}(\theta^{(t-1)})$
    $\gamma^{(t)} \leftarrow \gamma^{(t-1)} + \eta \Delta_\gamma$
    $\theta^{(t)} \leftarrow \mathrm{argmin}_{\theta \in \Theta} \left( g(\theta) + \left\langle \vec{\lambda}(\gamma^{(t)}), \vec{h}(\theta) \right\rangle \right)$
**end for**
$\theta^* \leftarrow$ a random variable that equals $\theta^{(t)}$ with probability $1/T$ for every $t \in [T]$
$\gamma^* \leftarrow$ a random variable that equals $\gamma^{(t)}$ with probability $1/T$ for every $t \in [T]$

---

**Algorithm 2** uses stochastic gradient descent and ascent to optimize the modeled Lagrangian of Equation 3: $\tilde{\mathcal{L}}(\theta, \gamma) = g(\theta) + \left\langle \vec{\lambda}(\gamma), \vec{h}(\theta) \right\rangle$. This is basically the algorithm that we use in our experiments (although often with a different first-order method than SGD). Because we do not assume convexity, this algorithm is *not* guaranteed to converge to a mixed Nash equilibrium, but it seems to work well in practice.

---

$\theta^{(0)} \leftarrow 0$    // or some other initial value ....
$\gamma^{(0)} \leftarrow 0$    // ....
**for** $t = 1$ **to** $T$ **do**
    $G_\theta \leftarrow$ a stochastic gradient of $g(\theta^{(t-1)})$
    $J_\theta \leftarrow$ a stochastic Jacobian of $\vec{h}(\theta^{(t-1)})$
    $\Delta_\theta \leftarrow G_\theta + J_\theta^T \vec{\lambda}(\gamma^{(t-1)})$
    $\theta^{(t)} \leftarrow \theta^{(t-1)} - \eta_\theta \Delta_\theta$
    $J_\gamma \leftarrow$ a stochastic Jacobian of $\vec{\lambda}(\gamma^{(t-1)})$
    $\Delta_\gamma \leftarrow J_\gamma^T \vec{h}(\theta^{(t-1)})$
    $\gamma^{(t)} \leftarrow \gamma^{(t-1)} + \eta_\gamma \Delta_\gamma$
**end for**

---

Adding Equations 11 and 12 together yields:

$$\sup_\gamma \frac{1}{T} \sum_{t=1}^{T} \tilde{\mathcal{L}}(\theta^{(t)}, \gamma) - \inf_{\theta \in \Theta} \frac{1}{T} \sum_{t=1}^{T} \tilde{\mathcal{L}}(\theta, \gamma^{(t)}) \leq \epsilon$$

$$\sup_\gamma \mathbb{E} \left[ g(\theta^*) + \left\langle \vec{\lambda}(\gamma), \vec{h}(\theta^*) \right\rangle \right] - \inf_{\theta \in \Theta} \mathbb{E} \left[ g(\theta) + \left\langle \vec{\lambda}(\gamma^*), \vec{h}(\theta) \right\rangle \right] \leq \epsilon$$

which is, aside from the $R$ scaling, exactly the requirement of Theorem 1 (Equation 5).

**"Practical" algorithm:** In our experiments, we use something closer to Algorithm 2, which is not guaranteed to converge to a mixed Nash equilibrium, but is easy to implement and works well in practice. This algorithm, however, leaves one major question unanswered: "how do we find stochastic gradients?". Typically, this will be done by sampling. The objective function will generally be an empirical average over a dataset, so we can find a stochastic gradient of $g$ by sampling a minibatch from the training set. When the constraints have the same form, the same approach can be used, and when the constraints are functions of individual examples (or queries), each row of the Jacobian can be calculated exactly without sampling.

Since our underlying assumption is that $m$ is extremely large, we would *also* like to sample the constraints. This, again, is straightforward: we sample a constraint $i \in [m]$ uniformly, find a

(stochastic) gradient of $h_i$, and construct a Jacobian matrix that is all-zero, except that the $i$th row is equal to this gradient, multiplied by $m$ (to account for the sampling probability). This approach could of course be improved upon (minibatching being the most obvious change, but different importance sampling strategies could also be beneficial), but this discussion shows that, in principle, each iteration can be made extremely computationally inexpensive.

Notice that, unlike Algorithm 1, in Algorithm 2 we do not construct and return uniform distributions over the iterates. While one could do this, there are numerous other options, as well: one could take the last iterate, or heuristically search for the "best" iterate, or even search for the best pair of random variables supported on the iterates (i.e. the "shrinking" procedure of Cotter et al. [15]).

# D   Experimental Details for Fairness over Intersectional Groups

As noted in Section 6.1, we seek to minimize the overall error rate, subject to the constraint $\text{error}(G_{t_1,t_2,t_3}) \leq \text{error}(\text{ALL}) + 0.01$, for all 1000 thresholds $(t_1, t_2, t_3)$, except those groups that contain less than 1% of the data. This results in a total of 822 constraints. Since some of the threshold combinations represent the same partitioning of the data, only 535 of the constraints are unique (i.e. are computed on distinct groups).

We use Algorithm 2 to solve the resulting constrained optimization problem, where at each step we sample a single threshold combination $(t_1, t_2, t_3)$ uniformly at random, and compute a stochastic gradient using the sampled constraint. We use Adagrad for the individual gradient updates, run the algorithm for a total $T = 10000$ gradient steps, with step-sizes $\eta_\theta = 0.1$ and $\eta_\gamma = 0.5$. After training, we use the "best iterate" heuristic of Cotter et al. [24] to pick a model that best trades-off between the objective and constraints. We train a linear classifier in all experiments and use hinge surrogates to approximate the objective and constraints. We implemented our method in TensorFlow using the open-source TensorFlow Constrained Optimization (TFCO) library [24][2].

We split the dataset into 2/3-rd for training and 1/3-rd for testing, and average the results over 5 such random train-test splits. We measure the violation in the constraint for group $G_{t_1,t_2,t_3}$ as: $\text{error}(G_{t_1,t_2,t_3}) - \text{error}(\text{ALL}) - 0.01$, where a negative violation value indicates that the constraint is satisfied. In Table 2, we report the error rates and the 95-th percentile constraint violation for imposing the constraints with five Lagrangian model architectures: (i) a *common* multiplier for all constraints, (ii) a linear model, (iii) a neural network with a single hidden layer of 50 nodes, (iv) a neural network with two hidden layers of 50 nodes each, and (v) a neural network with three hidden layers of 50 nodes each. Note that (iv) and (v) are over-parameterized models, i.e. have more parameters than the number of constraints. We compare our approach with an *unconstrained* baseline that optimizes error rate without constraints.

We ran experiments on a single machine with a 36-core Intel(R) Xeon(R) Gold 6154 Processor (3.00 GHz) and 191GB RAM. For this dataset, our approach incurs similar running times with the different Lagrange multiplier models. For example, with the linear multiplier model it takes 31.7 minutes, whereas with the 3-layer multiplier model it takes 33 minutes.

## D.1   Comparison with Kearns et al. [3]

As noted in Section 2, Kearns et al. [3] also consider the setting of intersectional fairness constraints, but impose constraints on subgroups defined by a class of membership functions over protected attributes. They then show how one can provably impose constraints on these subgroups when the class of membership functions has bounded VC dimension. On the other hand, we make no assumption on the given groups, but instead limit the flexibility of the Lagrange multiplier model used to impose constraints on them.

Our optimization strategy is also very different. Kearns et al. [3] propose having the $\lambda$- and $\theta$-players play Fictitious Play using an oracle to find the most-violated constraint at each step. Such an oracle, however, many not be efficient to implement for general constraint sets. So in our algorithm (Algorithm 2), we do not assume access to an oracle for picking the most-violated constraint, but instead sample a constraint uniformly at random, compute gradients for the sampled constraint, and perform stochastic gradient updates on $\gamma$ and $\theta$.

For completeness, we compare our results with the FairFictitiousPlay algorithm of Kearns et al. [3] (see Algorithm 3 in their paper). We impose equal error rate constraints on the same threshold-based groups considered in our original experiment, and use a brute-force search strategy to find the most-violated constraint. Note that such a brute-force oracle may not be efficient to implement when the number of protected attributes is very large. We run their algorithm for a total of 100 rounds of Fictitious Play, with 250 gradient steps to implement the cost-sensitive learner needed for the $\theta$-player's update step, and with the multiplier radius parameter $C$ chosen from the range $\{1.0, 5.0, 10.0\}$ to minimize the train error rate while satisfying the constraints.

As seen in Table 2, the proposed approach with a 3-hidden layer Lagrangian model is able to satisfy the constraints at least as well as the brute-force FairFictitiousPlay variant, while yielding a similar error rate. Moreover, the proposed approach is able to achieve this performance using a simple random sampling strategy to pick the constraint at each step, and without relying on an inefficient brute-force search to pick the maximally violated constraint.

# E  Experimental Details for Fairness with Noisy Group Memberships

The generation of noisy protected groups is the same as in Wang et al. [10]. In the experiments, we replace all the expectations in the objective and constraints with finite-sample empirical versions.

Let $f(x; \theta)$ be a binary classifier, where $f(x; \theta) > 0$ indicates a positive classification. Let $Y \in \{0, 1\}$ be a binary classification label. Recall that $M$ is a candidate true group assignment over a dataset $\mathcal{D}$: in finite samples, $M \in \mathcal{M} = \left\{ \{G_j^{(1)}\}_{i=1}^n, \{G_i^{(2)}\}_{i=1}^n, \ldots \right\}$, and $M_i = j$ implies that the $i$th example is assigned to group $j$.

To equalize true positive rates between groups with slack $\alpha$, we set

$$h_j(\theta, M) = \frac{\mathbb{E}_{x,y \sim \mathcal{D}}[\mathbf{1}\{f(x; \theta) > 0, y = 1\}]}{\mathbb{E}_{x,y \sim \mathcal{D}}[\mathbf{1}\{y = 1\}]} - \frac{\mathbb{E}_{x,y,M \sim \mathcal{D}}[\mathbf{1}\{f(x; \theta) > 0, y = 1, M = j\}]}{\mathbb{E}_{x,y,M \sim \mathcal{D}}[\mathbf{1}\{y = 1, M = j\}]} - \alpha.$$

We evaluate fairness violations $h_j(\theta, G)$ for the true group membership assignment $G$ on the test dataset. We do not directly use $G$ to train the binary classifier $f$. The only place we use $G$ in the training process is to estimate the marginal probabilities $P(G = i | \hat{G} = j)$ ahead of time on the training set, as done by Wang et al. [10]. These marginal probabilities are used to generate the candidate set $\mathcal{M}$.

We use Algorithm 2 to solve the resulting constrained optimization problem, where at each step we sample a group membership assignment $M$ uniformly at random, and compute a stochastic gradient using the sampled constraint. We train a linear classifier in all experiments and use hinge surrogates to approximate the objective and constraints. For the Lagrange multiplier, we use a linear model with a one-hot encoding $M$ as input. While this amounts to maintaining a total of $n$ parameters, one for each entry $M_i$, this is far fewer than the total number of constraints, which is exponential in $n$.

After training, we report results from the best iterate, where we define "best" as the iterate that achieves the lowest objective value [24], while also satisfying all constraints on a sample of 20 candidates from the candidate set $\mathcal{M}$.

For this experiment, we trained on a single machine with a 6-core Intel Xeon E5-1650 V3 Processor (3.50GHz) and 32GB RAM. The average training time over 5 trials for the proposed method was 13.36 minutes. The results from Wang et al. [10] were taken directly from the results reported in that paper, and were not retrained or reproduced in this paper. Their results were trained using a 4-core Intel Core i7-7700HQ CPU (2.80GHz) and 16GB RAM. The average training times over 5 trials are: DRO [10]: 5.56 minutes, and SoftAssign[10]: 5.52 minutes.

## E.1  Comparison to baseline with a single Lagrange multiplier per group

As a baseline to illustrate the effectiveness of the Lagrange multiplier model, we also trained using the same setup as Section 6.2, but instead of the linear Lagrange multiplier model described in Section 6.2, we simply update a single Lagrange multiplier $\lambda_j$ per group $j \in [k]$. We still resample the group assignments $M \in \mathcal{M}$ each epoch, but we associate a single Lagrange multiplier $\lambda_j$ with constraint $h_j(\theta, M)$. Results in Table 5 on the five different noise levels show that this single

Table 5: Equal opportunity constraints on Adult with different group noise levels in the training data (averaged over 10 trials). We report the test error rates (*lower* better) and test fairness violations evaluated with the *true* groups. A negative fairness violation indicates the constraints are satisfied.

| Noise | Per-group multiplier | |
| | Error rate | Violation |
| --- | --- | --- |
| 0.1 | $0.146 \pm 0.001$ | $0.008 \pm 0.030$ |
| 0.2 | $0.146 \pm 0.002$ | $0.040 \pm 0.015$ |
| 0.3 | $0.146 \pm 0.001$ | $0.057 \pm 0.018$ |
| 0.4 | $0.146 \pm 0.002$ | $0.068 \pm 0.021$ |
| 0.5 | $0.145 \pm 0.002$ | $0.068 \pm 0.008$ |

Lagrange multiplier per group achieved approximately the same error rate and true group violations as the unconstrained model. Thus, even though the group assignments $M \in \mathcal{M}$ were resampled each epoch, the shared Lagrange multiplier per group did not have the complexity to control the violations over all group assignments $M \in \mathcal{M}$.

### E.2 Relation to soft group assignments from Wang et al. [10]

We discuss the relation of the generated candidate groups with the soft group assignments proposed in Wang et al. [10]. We show that the generated samples are actually "hard" group assignments, which are in a subset of the soft group assignments. Therefore, the proposed algorithm in Section 6.2 solves an optimization problem in which the constraints are sampled from the feasible set of the soft group assignments.

The soft group assignments approach in Wang et al. [10] uses a function $w : [k] \times \{0,1\} \times \{0,1\} \times [k] \to [0,1]$ to estimate $P(G = j|\hat{y}, y, \hat{G} = k)$ by $w(j \mid \hat{y}, y, k)$, where $\hat{y} = \mathbf{1}(f(x; \theta) > 0)$. The soft group assignments approach in Wang et al. [10] then constrains $w$ in a feasible set $\mathcal{W}$ by enforcing the marginal probabilities estimated from auxiliary datasets:

$$\mathcal{W}(\theta) = \left\{ w : \begin{array}{l} \sum_{\hat{y}, y \in \{0,1\}} w(j|\hat{y},y,k) P(\hat{y}, y|\hat{G}=k) = P(G=j|\hat{G}=k), \\ \sum_{j=1}^{m} w(j|\hat{y},y,k) = 1, w(j|\hat{y},y,k) \geq 0 \quad \forall \hat{y}, y \in \{0,1\}, k \in \hat{\mathcal{G}} \end{array} \right\}. \tag{13}$$

The robust optimization problem takes the form:

$$\begin{aligned} \underset{\theta \in \Theta}{\text{minimize}} \quad & g(\theta) \\ \text{s.t. } \forall j \in [k]. \quad & \max_{w \in \mathcal{W}} h_j(\theta, w) \leq 0 \end{aligned}$$

where $g(\theta)$ represents the training loss, i.e. average error rates; $h_j(\theta) = \mathbb{E}[l_1(\theta, x, y)|G = j]$ where $l_1(\theta, x, y)$ represents some function to constrain over the true groups to ensure certain fairness constraints (see Wang et al. [10]).

Instead of solving the maximization as Wang et al. [10], the proposed algorithm in Section 6.2 solves this problem empirically by sampling from the candidate set of possible true group assignments. The candidate set $\mathcal{M}$ that we consider in Section 6.2 is a subset of $\mathcal{W}(\theta)$, where the function $w$ simply assigns a "hard" group assignment to each data point, i.e. $\{w : w \in \mathcal{W}, w(j \mid \hat{y}, y, k) = 1 \text{ for some } j \in [k]\}$. Any feasible $w$ in this subset corresponds to applying a candidate true group assignment to the data points.

While any feasible solution to the robust optimization problem would be guaranteed to satisfy the fairness constraints on the true groups, this is not necessarily true of the empirical procedure in Section 6.2 of sampling a different candidate from $\mathcal{M}$ every epoch to perform gradient updates.

## F  Experimental Details for Ranking Fairness with Per-query Constraints

By labeling documents with relevance score 3 and 4 as positive, we end up with approximately 12.5% positive documents overall. As the pairwise constraints are better captured when the positive and negative labels are balanced, we take a two-step approach to slightly adjust the ratio of positive and negative documents in the training set. We first generate two [-1,1] uniformly distributed random

Table 6: Pairwise ranking fairness with per-query constraints on W3C Expert (averaged over 5 trials). Constraint violation is measured as $|\text{err}_0 - \text{err}_1| - 0.3$ (*lower* is better).

| Method | Pairwise Error | | 90th Percentile Violation | |
|---|---|---|---|---|
| | Train | Test | Train | Test |
| Unconstrained | **0.4446** ± 0.0009 | **0.4704** ± 0.0758 | 0.2170 ± 0.0000 | 0.3252 ± 0.0247 |
| Narasimhan et al. [5] | 0.4528 ± 0.0018 | 0.4992 ± 0.0937 | 0.1932 ± 0.0205 | 0.2336 ± 0.0888 |
| Proposed | 0.4706 ± 0.0333 | 0.5384 ± 0.0844 | **0.1104** ± 0.0852 | **0.2082** ± 0.0499 |

numbers $(v_1, v_2)$ per query and discard a negative document from the query with the probability $(0.7 + 0.3 \cdot v_1) \cdot \frac{1}{2} \left(1 - v_2 \cdot \frac{m+1-2i}{m-1}\right)$ if it has the $i$th smallest QualityScore2 (feature 133) in the query. As a result, $v_1$ influences the number of negative documents, and $v_2$ controls the quality of the selected negative documents. After applying this step, we also filter out all queries that have either less than 10 positive documents or less than 10 negative documents. By doing so we prevent the calculation of cross-group error rates with overly small denominators. Finally, reporting the error rates, we assign a penalty of 0.5 for ties in the ranking.

We use Algorithm 2 to solve the constrained optimization problem, where at each step we receive a single query, and compute a stochastic gradient using the sampled constraint. We use Adagrad for the individual gradient updates, run the algorithm for a total $T = 100000$ gradient steps, with step-sizes $\eta_\theta = 0.05$ and $\eta_\gamma = 0.1$. After training, we use the "best iterate" heuristic of Cotter et al. [24] to pick a model that best trades-off between the objective and constraints. We train a ranking model with a single hidden layer of 128 nodes, and use hinge surrogates to approximate the objective and constraints. The Lagrange multiplier model is also a single-hidden-layer neural network with 128 nodes which takes the mean feature vector from a query as input. We implemented our method in TensorFlow using the open-source TensorFlow Constrained Optimization (TFCO) library [24].

We ran experiments on a single machine with a 36-core Intel(R) Xeon(R) Gold 6154 Processor (3.00 GHz) and 188GB RAM. The average training times over 5 trials for the unconstrained method is 3.13 hours, for the approach of Narasimhan et al. [5] is 4.43 hours, and for the proposed approach is 5.05 hours.

### F.1 Additional Experiments

We consider a similar ranking problem with a marginal equal opportunity fairness criterion defined as

$$\text{err}_i(q) = \mathbb{E}[\mathbf{1}\{f(X, q) < f(X', q)\}|Y > Y', G = i, Q = q]$$

and constraining $|\text{err}_0(q) - \text{err}_1(q)| \leq 0.3$ for all queries $q \in \mathcal{Q}$. We use the W3C Expert dataset studied in Zehlike and Castillo [21], Narasimhan et al. [5], which is a subset of the TREC 2005 enterprise track data. The dataset contains 60 topics, with 200 candidates per topic, where each candidate labeled as an expert or non-expert for the topic. The goal is to rank the candidates based on their expertise on a topic. We use the same features as Zehlike and Castillo [21] to represent how well each topic matches each candidate (a set of five aggregate features derived from word counts and tf-idf scores), and treat gender as the protected attribute. We learn a ranking model with one hidden layer of 8 nodes, and a multiplier model with one hidden layer of 8 nodes. We use 30 queries to learn both models using 20k gradient steps and a learning rate of 0.01, 15 queries to validate, then test the model performance on the left-out 15 queries. Table 6 summaries the pairwise error rates and the 90th percentile query level constraint violations. We see similar patterns as in Table 4 that the proposed method achieves the least constraint violation. However because of the small size of this dataset, we find that the standard deviations are generally higher, and the overall pairwise error on the test set is closer to 0.5. Because the proposed method more closely satisfies the per-query constraints it incurs a larger training error, and as a result a higher test error than the other methods.