[Reviews · NeurIPS 2020]

Review 1

Summary and Contributions: This paper contains an interesting idea. Suppose we wish to solve a linear or convex program with a very large number of constraints. One way to solve it is by finding a saddlepoint of the Lagrangian, say by running gradient descent over the Lagrangian weights and finding the primal optimizer at every round. But this requires maintaining a weight for every constraint, which is infeasible if there are many constraints. Instead, the authors propose that the "dual player" in this game use Lagrange multipliers that are functions of a much lower dimensional parameter space, and instead optimize over this lower dimensional parameter space. Its a natural thing to do. The authors prove a theorem justifying the approach under very strong assumptions (essentially, if one assumes that the parameterized family of Lagrange multipliers is rich enough to penalize any constraint violation at any realization of the primal variables, then finding a saddlepoint of the parameterized minmax problem solves the original constrained optimization problem). The theorem itself isn't so interesting because it essentially bakes in its conclusion into its assumptions, but the authors have some promising experiments showing the feasibility of their approach. =======AFTER RESPONSE====== Thanks for the response and to pointing out the experimental comparison to [3] in the Appendix. I agree this should be in the body; I'm raising my score. I would suggest framing the comparison to [3] and other related work somewhat differently. Right now, it is framed as (paraphrasing) "[3] assumes the constraints have bounded VC dimension, whereas we assume the dual solution has a low dimensional structure". This isn't right --- the VC-dimension assumptions in [3] are used only for generalization claims, and aren't important for optimization. Rather, what is used in [3] and many other settings in which there are exponentially many constraints is the ability to optimize to -find- a substantially violated constraint whenever one exists. It is true that this isn't always possible, and papers like [3] use heuristics to implement this step. In comparison, in practice, the theory for the approach proposed here relies on the (heuristic) assumption that the dual solution has some implicit low dimensional structure. Perhaps this is sometimes true even when optimization oracles are hard to implement --- your empirical comparison provides evidence that your method can be useful. But the paper will be much more correctly positioned if you start by explaining that there are a number of methods for solving heavily constrained convex programs efficiently already (even when the constraints are not known up front), but that they require dual optimization oracles, and you are investigating what can be done under a different set of assumptions, rather than doing something different in kind.

Strengths: The idea is very nice; obvious in hindsight, as the best ideas are. This likely leads to a practical approach to constrained optimization in highly constrained problems.

Weaknesses: The authors don't really compare to prior work, which includes other ways to solve highly constrained optimization problems, often with guarantees that do not require assumptions. For example, Kearns et al [3] cited by the authors not only proposes a fairness motivated problem with many constraints, but gives a way to solve it. In particular, they show how to run a no-regret algorithm (follow the perturbed leader) over the primal variables, which allows the "dual" player to best respond. This means that just as in this paper, it is not necessary for the constraints to be enumerable or even known ahead of time --- all that is required to run the algorithm is that violated constraints can be identified as they arise. Since it is finding a saddlepoint of the original Lagrangian, nothing like the present paper's Assumption 2 are necessary. So when is the approach in this paper preferable to the approach in Kearns et al? I imagine that there are circumstances when it is --- but the paper doesn't even attempt to make the comparison.

Correctness: Yes

Clarity: Yes

Relation to Prior Work: See above; there are existing approaches to dealing with large/unknown numbers of constraints, and the paper does not clearly position itself or explain when it yields an improvement. The authors also oddly attribute the technique of solving a linear program by finding the equilibrium of the Lagrangian game to Agarwal et al [2018], but this is a very old technique, dating back at least to Plotkin/Shmoys/Tardos '91 (although they used slightly different terminology). See the 2008 Arora et al. survey "The Multiplicative Weights Update Method: A Meta-Algorithm and Applications" for a treatment in more familiar language.

Reproducibility: Yes

Additional Feedback:


Review 2

Summary and Contributions: This paper investigates the optimization problems with many constraints, which appear in machine learning applications such as ranking fairness or fairness over intersectional groups. In these cases, the standard approach of optimizing a Lagrangian while maintaining one Lagrange multiplier per constraint may no longer be practical. The authors associate a feature vector with each constraint, and to learn a “multiplier model” that maps each such vector to the corresponding Lagrange multiplier. They prove optimality, feasibility, and generalization guarantees under certain assumptions on the flexibility of the multiplier model, and empirically demonstrate that their method is effective in real-world case studies.

Strengths: I like the idea of dimension reduction for Lagrange multipliers, which is novel in constrained optimization. The theoretical guarantees for optimality, feasibility, and generalization are interesting. The empirical results show that this idea works in real-world applications.

Weaknesses: As I know, this is the first paper that applies dimension reduction to handle Lagrange multipliers. But I doubt that the dimension reduction idea should have appeared in other areas before. I would like to see a comparison with similar approaches and an explanation of the novelty. Another question is about the empirical results. As mentioned in the abstract, one contribution is to show their method is effective. But I do not see a comparison of the running time between their method and baselines. -- Thank you for the author response. I think it addresses my questions.

Correctness: The claims, method, empirical methodology are correct.

Clarity: The paper is well written. A typo in Line 203, page 5. "in such in such a way".

Relation to Prior Work: Yes, the paper discusses the novelty compared to previous contributions.

Reproducibility: Yes

Additional Feedback:


Review 3

Summary and Contributions: The paper tackles the issue of solving large-scale constrained optimization problems using primal-dual methods. It addresses the computational issue of updating the dual variables, which in certain problems can be as many as entries in the dataset. To do so, they replace the high-dimensional dual variables by a lower dimensional representation. ======================================================== EDIT: I appreciate the answers from the authors. While I do not agree with their take on random models, I understand that this has become an accepted "hack" to deal with non-convexity. It doesn't detract from their paper, but it is a limitation that I believe is larger than the extent of its discussion make it look. The title, as I mentioned, is another minor point. I expected something else when I read it, namely, that the constraints would be tight or that the problem would be close to infeasible. That is clearly not the case as the work actually chooses to not even enforce the constraints (except in aggregate). That last point, I maintain, is one that is crucial (and perhaps a good reason to modify the title, abstract, and contributions). If a problem is posed with pointwise constraints it is because these constraints are necessary. Currently, the manuscript poses optimization problems with pointwise constraints, but then proceeds to solve a different problem (one where the constraints are aggregated). I believe this is a major issue: the algorithm effectively solve a different problem than the one that is posed. In practical safety applications where constraints denote contingencies, problems can have an extremely large number of constraints (in the tens of thousands) and yet they each need to be individually satisfied. Solutions to semi-infinite programs (infinite number of constraints) are not feasible in aggregate. While the idea put forward in the paper is interesting and worth exploring, it currently only provides weak guarantees at best. Nevertheless, I still believe the paper is good enough for publication and hope that the authors will take the time to properly address these issues in the camera-ready.

Strengths: The issue of solving problems with a large number of constraints is an important one and the learning approach proposed in the text is an interesting contribution. The numerical examples provided serve well to illustrate the advantages of the method and its potential application in ML.

Weaknesses: While the idea of updating a model for the dual variables is interesting, several aspects of the new solution need to be explored more in-depth in the manuscript. For instance, while the number of dual variables is the same as the number of constraints, their updates are considerably simpler. In other words, while the dual ascent can be a very high dimensional problem, its gradient is very simple to compute (since it is simply the constraint violations). So the updates are quite simple. While memory remains an issue, there is an increase in complexity in the update that needs to be balanced by appropriately selecting the dual variables model. The guarantees given in the paper are also somewhat vacuous. The use of the aggregation function Phi makes it so that the algorithm can only be guaranteed to provide approximate soft constraint satisfactions (e.g., averaging the constraints) instead of an approximate pointwise guarantee (e.g., approximately meeting each of the constraints individually). For applications such as fairness, this can be a serious issue since it is likely for a few constraints to be very hard to meet while others are trivial. While the generalization guarantees are more interesting they suffer from a similar issue: the guarantees are not provided for the original constrained problem, but for another one where the constraints are aggregated. This point and these discussions need to be made and carried out in considerable more details in the revised versions of the manuscript. Finally, the guarantees given in the paper are for randomized models (mainly due to the non-convexity assumption on the objective and constraints). While these are theoretically interesting, it considerably limits their applicability in practical scenarios. A discussion of these limitations and possible solutions should be included in future versions of the paper. Minor issues - The title of the paper does not represent well the idea. Perhaps something with "large scale" would be better suited. "Heavily" gives the idea that the constraints are somehow stronger that hard constraints.

Correctness: The paper appears to be correct.

Clarity: The paper is well-written and organized. The motivation and interpretation of the aggregation function Phi could be improved. As it is, the burden lies heavily on the reader to understand why it is used and how it impacts the contributions of the paper.

Relation to Prior Work: Yes

Reproducibility: Yes

Additional Feedback:

[Author Response · NeurIPS 2020]

We thank the reviewers for the detailed feedback. R1 and R2 ask for a comparison to a prior method. R3's main comments are about the theoretical results. We've addressed all of this below.

**(R1 & R2) Comparison to Kearns et al. in Appendix D.1:** We provide a comparison to the prior method by Kearns et al. (2018) [3] in Appendix D.1, and will move it to the main text. We'll also expand the discussion we have about this method in the related work section. Kearns et al. consider an intersectional fairness problem similar to ours, but assume that the subgroups are defined by a class of functions with finite capacity. In contrast, we make no assumption on the given groups, but instead limit the flexibility of the Lagrange multiplier model we use to enforce constraints on them. We are thus able to handle a more general set of constraints, including those for ranking fairness.

More importantly, Kearns et al. assume access to an oracle to find the maximally-violated constraint at each step of their optimization. This assumption can be unrealistic, e.g. with the ranking fairness problems we consider, where we have one constraint for each incoming query, and the oracle would need to solve a maximization problem over *all* queries seen so far, or worse, over all realizable queries (this may be impractical without additional strong assumptions). Our optimization strategy is more practical and computes a simple gradient update on the Lagrangian model for each sampled query.

Table 4 in the appendix contains an empirical comparison to Kearns et al. on the intersectional fairness task (Sec 6.1). With a 3-layer multiplier model, we were able to achieve a similar test error and 95-th percentile constraint violation as their method, which we implemented with a brute-force search to find the max-violated constraint (Proposed: 0.22 [0.03], Kearns et al: 0.22 [0.04]). We aren't aware of a straight-forward way to implement Kearns et al. for the ranking fairness experiments.

**(R1 & R3) Guarantees are on aggregated constraints:** When there are a small number of constraints, it only makes sense that we should require that they all hold simultaneously. However, in the heavily-contrained setting that we consider, in which there can even be a potentially infinite number of constraints, expecting a model to satisfy all the constraints may not be reasonable, and therefore an alternate approach must be taken. This is the role of the $\Phi$ function: it tells us how we should measure the magnitudes of the constraint violations. The reviewers' main concern, we believe, is that there is a close correspondence between the function class chosen for the Lagrange multiplier model, and the (potentially unknown) $\Phi$. It might be helpful to think of it this way: for every Lagrange multiplier model function class, there exist many possible choices of $\Phi$ (needed only by the theory, not the algorithm), each of which describes the sort of infeasibility and generalization guarantees that we can achieve. For some simple function classes, a convenient $\Phi$ is known (Table 1), while for more complicated function classes, it is not, but can still be reasoned about formally.

We also emphasize that we *do not* make a strong assumption about the constraints. While this allows us to handle general constraint sets, it restricts our ability to provide stronger theoretical guarantees.

**(R3) Stochastic classifiers:** In theory, stochastic classifiers are difficult to avoid for *non-convex* Lagrangian optimization, since the minimax theorem doesn't hold, so one must seek a *mixed* equilibrium (see e.g. ref. [11, 12] for prior use of stochastic classifiers for constrained problems). In practice, however, deterministically taking the last iterate usually (but not always) works, and if it doesn't, one can convert the stochastic classifier to a deterministic one (Cotter et al., 2019).[1] We'll add a brief discussion and citation of Cotter et al. to the paper.

**(R2) Run-times:** All experiments were run with TensorFlow. For the intersectional fairness experiment (Sec 6.1), our method took 31.7 mins with a linear multiplier model, and 33 mins with a 3-layer multiplier model. For the ranking fairness experiment on the larger MSLR-WEB10K data (Sec 6.3), our method took on average 6.8 hrs to train, the baseline constrained method took 6.2 hrs, and the method with no constraints took 4.3 hrs. We'll report these numbers, and also include the run-times for the experiments in Sec 6.2, along with details about the CPU configurations used.

**(R3) Complexity of multiplier model:** We expect that on a real-world problem, as one adds constraints, they will tend to become increasingly redundant. Our intuition is that the Lagrange multiplier model can *learn* these redundancies, and therefore cope with much larger constraint sets than one might expect based on its complexity alone. For example, in our robust fairness experiment in Sec 6.2, the number of constraints is exponential in the data size (one for each grouping of the data), but the multiplier model we used needed far fewer parameters.

**(R3) Title:** Thank you for the suggestion—we'll definitely consider other titles.

## Footnotes

[1]A. Cotter, H. Narasimhan, and M. Gupta. On making stochastic classifiers deterministic. In *NeurIPS*, 2019.


[Meta-Review · NeurIPS 2020]

This paper proposes a new way of handling a large number of constraints in constraint optimization problems by compressing the (large number) of lagrange multipliers into a lower dimensional vector. This type of optimization is motivated by learning under fairness constraints, where a model may need to satisfy various conditions for various subgroups of the population. The authors prove optimality and generalization guarantees under certain assumptions and establish the practicality of their algorithm empirically. The submission received three knowledgeable and enthusiastic reviews recommending it as a clear accepeptance to neurips. Several reviewers however marked that some parts (abstract, intro, discussion of related work) are framed in a misleading way and gave recommendations for a more suitable presentation. The authors are asked to take these into account when preparing the camera ready version of their paper.